# Miniaturization of Respiratory Measurement System in Artificial Ventilator for Small Animal Experiments to Reduce Dead Space and Its Application to Lung Elasticity Evaluation

**DOI:** 10.3390/s21155123

**Published:** 2021-07-28

**Authors:** Homare Yoshida, Yoshihiro Hasegawa, Miyoko Matsushima, Tomoshi Sugiyama, Tsutomu Kawabe, Mitsuhiro Shikida

**Affiliations:** 1Department of Biomedical Information Sciences, Hiroshima City University, Hiroshima 731-3194, Japan; hasegawa@hiroshima-cu.ac.jp (Y.H.); shikida@hiroshima-cu.ac.jp (M.S.); 2Division of Host Defense Sciences, Omics Health Sciences, Department of Integrated Health Sciences, Graduate School of Medicine, Nagoya University, Nagoya 461-0047, Japan; matsu@met.nagoya-u.ac.jp (M.M.); kawabe@met.nagoya-u.ac.jp (T.K.); 3Department of Thoracic Surgery, Graduate School of Medicine, Nagoya University, Nagoya 466-8560, Japan; t-sugiyama@med.nagoya-u.ac.jp

**Keywords:** ventilator, barotrauma, dead space, miniaturization, lung elasticity evaluation

## Abstract

A respiratory measurement system composed of pressure and airflow sensors was introduced to precisely control the respiratory condition during animal experiments. The flow sensor was a hot-wire thermal airflow meter with a directional detection and airflow temperature change compensation function based on MEMS technology, and the pressure sensor was a commercially available one also produced by MEMS. The artificial dead space in the system was minimized to the value of 0.11 mL by integrating the two sensors on the same plate (26.0 mm × 15.0 mm). A balloon made of a silicone resin with a hardness of A30 was utilized as the simulated lung system and applied to the elasticity evaluation of the respiratory system in a living rat. The inside of the respiratory system was normally pressurized without damage, and we confirmed that the developed system was able to evaluate the elasticity of the lung tissue in the rat by using the pressure value obtained at the quasi-static conditions in the case of the ventilation in the animal experiments.

## 1. Introduction

Ventilators [1,2,3,4] are used in the medical field to assist with the maintenance of breathing as a vital activity, and they are also applied to research on the respiratory system in animal experiments. Generally, the ventilator for small experimental animals consists of a mechanically simple piston structure, and it generates a sinusoidal shaped respiration airflow. As a result, the respiration airflow conditions, such as the tidal airflow volume and the frequency, are determined prior to the experiments, and these values are not normally changed during the experiment [5,6]. However, the respiratory system properties, e.g., tissue elasticity, sometimes change during the experiments depending on what kind of experiment it is. This leads to various problems when the respiration airflow conditions do not follow along with the change of the respiratory system properties. For example, the lung tissue could be physically damaged when a positive excessive pressure airflow from the ventilator is supplied [7,8,9,10,11,12,13,14,15,16]. Conversely, choking may occur if there is a respiration airflow volume shortage caused by a poorly fitted tube in the ventilator system. In addition, uneven ventilations between the alveoli will be unresolved by the plateau effect [17] (stagnation) when the sinusoidal-waved respiration airflow from the ventilator is applied.

The first problem abovementioned, the ventilator barotrauma, is related to the increment of the pressure value inside the respiratory system, and the second one is caused by the change of the airflow volume by the ventilation. To solve these problems, two measurements devices are required: a sensor for the pressure and another for the airflow volume. Pressure and airflow volume measurements devices for the ventilator have recently been developed as options for monitoring the respiratory system properties during experiments. However, these devices are generally large, heavy, and expensive.

From the technical aspect of the sensors developments, various types of physical sensors, for example pressure [18,19], airflow [20,21], shear stress [22,23] acceleration [24,25,26], gyro [27,28], and force [29,30,31], have been miniaturized and integrated onto a chip for increasing the functional capability in a system by Micro Electro Mechanical Systems (MEMS) technologies. Both pressure and airflow sensors have been developed from the beginning of MEMS, and the small-sized sensors with a size of a few mm are now applied in automobile, health monitoring, and air-conditioning systems [32,33]. Thus, to overcome above-stated problems in the usage of the ventilator in the animal experiments, the miniaturized thermal flow and pressure sensors were applied to produce a cheap and light respiration monitoring system for ventilators in our previous studies [34,35]. A small balloon having a similar volume to that of the respiratory system of small experimental animals was used as the simulated lung, and it was confirmed that the developed respiration monitoring system could statically detect values of both the pressure and airflow volume inside the balloon supplied from the syringe pump, after which the balloon elasticity could be derived from these values. Additionally, a measurement method based on the quasi-static condition was introduced to derive the balloon elasticity in the case of the sinusoidal airflow condition supplied by the ventilator. The developed respiration monitoring system was finally applied to a rat, and values of both the pressure and the airflow volume in the respiratory system were successfully detected.

However, each pressure and airflow sensor element was connected by a tube a few dozen mm long, which extended the artificial airway. Thus, the previous respiration monitoring system has a problem in that its dead space increment [36,37,38] by the tube connection leads to insufficient aeration. Thus, in this study, we newly integrated both the pressure and airflow sensors on a chip substrate to minimize the artificial dead space. We applied MEMS technology [39,40,41,42] to manufacture the integrated system and tackled the evaluation of lung tissue elasticity as one of the respiratory system properties in rats under the sinusoidal airflow condition in the ventilator.

## 2. Respiratory Measurement System

### 2.1. Dead Space at Lung System in Animal

The normal tidal volume (airflow volume per cycle) is usually set to 6.0 mL/kg when small animals (mammals) are used in experiments, about one-third of which are called the physiological dead space volume. Therefore, the dead space ventilation ratio (V_D_/V_T_) between the tidal volume (V_T_) and the physiological dead space volume (V_D_) becomes about 0.3 in the animals. In the case of animal experiments, the ventilator and respiratory monitoring systems are connected to the airway of the experimental animals by means of a flexible tube of a certain length. This means the dead space ventilation ratio increases according to amount of tube space working as the artificial dead space, and insufficient aeration will occur when the ratio becomes high.

The average body weight of the rats used in our study was roughly 0.3 kg, so the values of the tidal volume and the physiological dead space became 1.8 mL and 0.6 mL, respectively. In the previous study, the artificial dead space caused by the tubing in the respiratory measurement system was 0.47 mL, and thus the total dead space in the respiration became 1.07 mL, corresponding to 60% of the tidal volume. As a result, insufficient aeration was caused when the experiment reached the 2–3 min mark. To resolve this issue, we integrated the pressure and airflow sensors on a chip in order to decrease the artificial dead space. The detailed specifications are discussed in the following section.

### 2.2. Structure and Fabrication

The structure of the respiratory measurement system for the animal ventilation is shown in Figure 1. It had a rectangular outer shape of 26.0 mm × 15.0 mm × 3.5 mm, and we sandwiched the sensor-device area with 1.0-mm-thick glass plates. Three layered polystyrene plates with a thickness of 0.5 mm and sensor film made of polyimide with a thickness of 7.5 µm were used to produce the flow channel and the airflow sensor, respectively. Two-hole patterns were formed on the bottom-layered plate for thermal isolation from the sensor film and the electrical feedthrough from the sensor element. Two plates were applied as the middle-layered plates and assembled on the film sensor with a gap of 6.5 mm, corresponding to the flow channel width. The flow channel was formed by placing the top-layered plate between them. Thus, the thickness of the middle layer corresponded to the height of the flow channel.

A micro-machined thermal sensor is frequently applied in airflow measurement, especially in the small flow rate region, because it can provide a high sensitivity measurement with a simple structure. Three different types of principles, a hot-wire, a calorimetric, and a time-of-flight airflow meter were used as the thermal sensors [43,44,45]. A hot-wire thermal airflow meter with a directional detection and airflow temperature change compensation function [46,47] based on MEMS technology was used in this study, as it has a large dynamic range in the airflow rate measurement. The heater worked as the hot-wire thermal airflow sensor, the meander-shaped metal patterns worked as the airflow direction sensor, and the airflow temperature compensation sensor was fabricated in the thin polyimide film with a thickness of 7.5 µm to reduce the thermal capacity. A photolithography, a thin metal film deposition by a sputtering, and a lift-off process were used in the fabrication. Two layered metals composed of Cr and Au were used as the sensor materials. Au was used as the sensor material, and Cr was applied as the adhesion layer between the polyimide and the Au layer. The sensor film applied in this study is shown Figure 2a (see ref. [47] for a detailed fabrication process). The sensor film was elastically bent to fit on the inside tube surface, as in the case of the previous study [34,35], however the planar one was simply placed on the polystyrene plate in this study, as shown in Figure 1. As stated above, the width and height of the flow channel at the airflow sensing area were 6.5 mm and 0.5 mm, respectively, so the airflow distribution became a plane Poiseuille flow at the airflow sensing area. The theoretical Reynolds number, expressed by a kinetic viscosity (m^2^/s), an average velocity (m/s), and a characteristic linear dimension (m), was estimated to confirm the flow condition in the system. The hydraulic diameter was used as the value of the characteristic linear dimension in the case of the rectangular-shaped flow channel applied in the airflow sensor area, and it approximately became double the flow channel’s height due to its wide shape. The number of the order of hundreds was derived from the following experimental conditions: the kinetic viscosity value of air: 1.5 × 10^−5^ m^2^/s at 20 °C, the average velocity 0.77 m/s in the case of the tidal volume of 5.0 mL driven by 0.5 Hz sinusoidal-shaped oscillating airflow, and the hydraulic dimeter 1.0 mm (double the flow channel height). Thus, we confirmed that the air flow in the system becomes the laminar flow in our experimental conditions.

The cavity with a height of 0.5 mm on the backside of the sensor film was formed for the thermal isolation, minimizing the thermal capacity. Elastic tubes with a 2.0 mm inner diameter deformed to the ellipse at the cross-sectional direction were connected to the rectangular shaped flow channel from the device plate. They were sandwiched by the solidified dimethylpolysiloxane (PDMS) sheets in the lateral direction, and liquid PDMS was poured into the gap between them as the sealing material. One part of the side surface body of one tube was opened to form a hole 0.1–0.2 mm in diameter, and another tube was placed in the opened hole to access the pressure sensor.

For the pressure measurement, a commercial pressure sensor (Honeywell gauge pressure type ABPDLNN030PGAA5, Charlotte, NC, USA) produced by MEMS technology was used in this study (see Figure 2b), because the small-sized sensor with a high sensitivity is now easily available in the market. Two types MEMS pressure sensors, absolute- and gauge-pressures, are generally provided. The former and the latter express the pressure values based on the vacuum and atmospheric pressure, respectively. Conversely, the needed specifications of the pressure sensor were decided as follows from the previous experiments:(1)Variation of pressure value; the pressure variation insides of the balloon and the respiratory in the rat caused by the airflow supply are assumed up to 10 kPa and 1.0 kPa, respectively.(2)Resolution; the pressure variation becomes less than 100 Pa under the small airflow volume condition in the case of the rat. Thus, we assumed that a few Pa of resolution was needed in the following animal experiments.

On another front, the atmospheric pressure value is about 101 kPa, and thus the sensor has to detect the pressure variation from less than 100 Pa to 1.0 kPa with a few Pa resolutions in the case of the animal experiments. However, the atmospheric pressure value fluctuates according to the length of time and, for example, its value changed 140 Pa roughly during the previous animal experiments. Thus, the gauge-type pressure sensor was chosen in this study to eliminate the atmospheric pressure value fluctuation in the pressure measurements. The outer shape of the sensor having a single axial barbless port was also considered to facilitate its assembly into the system. The outer diameter and the length at the port were 2.7 mm and 4.8 mm, respectively. The sensor substrate and the weight were 8.0 mm × 11.0 mm and 0.67 g, respectively. The sensor was operated by the supply voltage of 5.0 V.

The dead space in our newly developed respiratory measurement system became 0.11 mL thanks to the integration of the sensor devices, which was reduced to 23% compared with the devices used in the previous study. The absolute value of 0.11 mL was relatively small to the physiological dead space of 0.6 mL in the case of the rat, which demonstrates that our respiratory measurement system could improve the air ventilation in animal experiments where the breathing is controlled by an artificial ventilator.

### 2.3. Calibration Method

A hot-wire thermal airflow sensor is operated by a constant-driving-voltage or constant-temperature-circuits, and they are controlled by an open- and a closed-loop, respectively. To shorten the response time, the thermal airflow sensor was operated at 76.5 °C by a constant temperature circuit, because it is applied to the oscillating airflow measurement by the ventilator. The experimental setup used for the calibration is shown in Figure 3. A compressed gas cylinder was applied as an air source, and a commercially available mass flow controller (KOFLOC Model 3200, Kojima Instruments Inc., Kyoto, Japan) was used as a reference to calibrate the fabricated airflow sensor. A three-port solenoid valve was placed in between the mass flow controller and the airflow sensor for the evaluating the following response properties. The constant temperature circuit and the solenoid valve were operated by the function generators.

The calibration curve showing the relationship between the airflow volume rate and the sensor output under both the forward and backward directions was obtained due to the oscillating airflow supply by the ventilator, as shown in Figure 4. The calibration range was empirically determined to be 0–2000 ccm based on past small-animal experiments. The calibration curve in the case of the hot-wire thermal airflow sensor operated by the temperature constant circuit is theoretically expressed by King’s equation [48], but the experimental data at the low flow rate region often scattered from the equation. King’s equation derived from the forced convection from the heater to the airflow, and therefore it normally fits with the experimental results obtained under the high-flow-rate region, as the forced convection dominates the heat flow in this range. Conversely, it does not fit at the low flow rate region because the free convection dominates the heat flow phenomena in this case. Therefore, an approximate polynomial equation with six order was applied as the calibration curve in this study. The assembly of the pressure sensor does not significantly affect dead space, so we used the commercially available miniaturized dip-type pressure sensor. The measurement range and the sensitivity were 0–206 kPa and 0.038 mV/Pa, respectively. The obtained sensor output voltage was translated into the pressure value by referring to the data sheet.

Additionally, the response time of both the airflow flow and the pressure sensors have to be shortened sufficiently to follow the airflow rate and the direction changes in oscillating airflow measurements supplied by the ventilator. The response property of the fabricated airflow sensor was experimentally examined. The step-shaped waveform was conducted as the airflow input, and thus it was evaluated by a step response. The three-port valve was used to generate the step-shaped airflow waveform (see Figure 3). The response properties under both the forward and the backward flow directions were evaluated in the same way to the above-stated flow detection characteristics.

The response waveforms obtained by the step airflow input at the airflow rate of 300 ccm are shown in Figure 5. The response time value was defined as the time period in which the sensor output reached to 90% of the steady state, and those values were derived from the response waveforms. The obtained response time values under four different conditions, raising and falling under the forward flow and raising and falling under the backward flow, were 63 ms, 65 ms, 64 ms, and 51 ms, respectively. In our study, the targeted respiratory frequency of a rat was a few Hz corresponding to several hundred ms in the cycle time, and thus we concluded that the produced airflow sensor had a sufficient response speed following the oscillating airflow supplied by the ventilator.

As stated above, we applied the commercially available miniaturized dip-type pressure sensor, which has a sufficiently high response speed of 1 ms for oscillating airflow measurement by the ventilator.

## 3. Experiments

### 3.1. Outline of Experiments

The experimental setup is shown in Figure 6. A balloon made of a silicone resin with a hardness of A30 was first applied as the artificial lung system for the evaluation of the developed respiratory measurement system, prior to applying it to living respiratory evaluation. We used a sphere-shaped balloon with an inner volume of 0.68 mL, which is almost the same as the respiratory system in small experimental animals. The inner diameter and the thickness were 1.0 mm and 0.3 mm, respectively. It was attached at one end of the respiratory measurement system. The elasticity of the balloon was evaluated by supplying the airflow with a syringe and the ventilator (Harvard mouse ventilator (model 683), Holliston, MA, USA).

After that, the airway of the rat (weight: 326 g) was connected to the end of the developed respiratory measurement system, and we investigated whether our system could detect the elasticity of the living rat. The supplied airflow volume was gradually increased, and we alternately evaluated the results of the syringe and the ventilator airflow supply so that we could compare them. The animal experiments were conducted in accordance with the Nagoya University Animal Experiment Regulations with the approval of the Animal Ethics Committee (Approved number: 20021, Date: 16 March 2020). The detailed experimental procedure and results are discussed in the following sections.

### 3.2. Balloon Elasticity Evaluation

The elasticity of the balloon was expressed by the ratio between the airflow volume and the pressure value inside the balloon and, therefore, the relationship between them was evaluated.

First, we used the no-flow condition (static condition: Pressure was statically balanced) to precisely evaluate the pressure value inside the balloon. The syringe was connected to the other end of the respiratory measurement system, and the airflow was supplied from the syringe to the balloon from 0.2 mL to 0.8 mL in 0.2-mL increments. After the airflow supply, the three-way stopcock was switched and closed to make a statically pressure balanced condition in the system, and the pressure value was measured (static condition: no-flow and constant pressure). Then, the three-way stopcock was slowly opened and the pressurized air inside the balloon was exhausted to the atmosphere, and the airflow volume stored in the balloon was detected by the airflow sensor. The typical measured airflow and pressure waveforms when the airflow volumes were 0.2 mL and 0.6 mL are shown in Figure 7. The pressure value became a constant after the airflow supply by the syringe, as shown in Figure 7, and thus we confirmed that the pressure inside the balloon was statically balanced.

The relationship between the airflow volume and the pressure value in the balloon obtained under the statically pressure balanced condition is plotted by the open dots in Figure 8. The gradient of the pressure vs. flow volume curve shows the elasticity, and the pressure non-linearly increased with the increase of the airflow volume in the case of the balloon. Its value rapidly increased at the pressure value of 107 kPa. Thus, we concluded that the balloon used in this experiment had the following characteristics:(1)The balloon elasticity was almost constant in the region of the small balloon deformation by the airflow inflating.(2)The balloon elasticity largely changed in the large deformation region when the balloon was inflated over the pressure of 107 kPa.

Conversely, the sinusoidal-shaped oscillating airflow was basically supplied by the ventilator in the case of the animal experiments, and thus the balloon elasticity evaluation under the oscillating airflow was confirmed. The ventilator was connected to the system, and the sinusoidal-shaped oscillating airflow at a fixed frequency of 1.0 Hz was applied. The airflow volume (tidal airflow volume) was changed from 0.25 mL to 1.75 mL in 0.25-mL increments by switching the control valve in the ventilator. The air was cyclically supplied to the balloon from the ventilator, and the supplied airflow was then exhausted to outside of the atmosphere. The obtained typical airflow and pressure waveforms when the airflow volumes were 1.0 mL and 1.5 mL in the case of the ventilator are shown in Figure 9. As shown in the airflow and the pressure waveforms, the sinusoidal-shaped airflow was supplied by the motion of the piston in the ventilator, and the pressure value inside the balloon increased according to the airflow in the case of the airflow supply. Conversely, the pressure value became to zero in a moment due to the pressurized air relief to the atmosphere in the case of the air exhaust.

The airflow became zero when the airflow direction was changed in the case of the sinusoidal-shaped oscillating airflow supply, as shown in Figure 9. Thus, the pressure value, in which the airflow direction was changed from the supply to the exhaust, was used as the value under the quasi-static condition in the case of the oscillating airflow supply [18,19]. The relationship between the airflow volume and the pressure value in the balloon obtained under the sinusoidal-shaped oscillating airflow condition is plotted by the solid black circle in Figure 8. The relationships obtained under both the static and quasi-static conditions were coincident, which demonstrates that we were able to evaluate the balloon elasticity by using the pressure value under the quasi-static condition, even if we use the sinusoidal-shaped oscillating airflow as the air ventilation.

### 3.3. Elasticity Evaluation of Lung Tissue in Living Rat

The respiratory measurement system was placed between the airway extracted by tracheostomy from the rat’s pharynx and the syringe or the ventilator (see Figure 6). Figure 10 shows the photograph of the animal experiment when the ventilator was applied. The developed respiratory measurement system was directly connected to the airway of the rat. In the previous study, insufficient aeration was caused when the experiment reached the 2–3 min mark. Thus, the system was frequently disconnected from the airway to resume the breathing in the measurements. However, thanks to the reduction of the dead space in the newly developed system by the sensor integration onto the chip, we could keep its connection with the airway up to the end of the following elasticity evaluation of the lung tissue in the rat. The biological reactions caused by the insufficient aeration did not occur even if the developed system was connected during the experiments in this study.

The tidal volume in the case of the rat (weight: 0.2–0.4 kg) corresponds to almost 2.0 mL, so we set the maximum ventilation volume to 3.0 mL to avoid damage to the living body and the respiratory system. Using the same approach as the balloon experiments above, we evaluated the elasticity of the lung tissue in the living rat under both the static condition for the syringe airflow case and the quasi-static one for the sinusoidal-shaped oscillating airflow.

In the case of the static condition, the syringe was connected to the respiratory measurement system, and the amount of airflow volume from the syringe was increased in 0.5-mL increments. Spontaneous breathing causes backing, which makes it difficult to measure both the pressure and the airflow volume, so we performed the experiments under deep anesthesia (coma condition) to avoid spontaneous breathing. The ventilator assistance was incorporated to stabilize the physical condition after each measurement interval. The airflow and pressure waveforms when the airflow volumes were 2.0 mL, 2.5 mL, and 3.0 mL are shown in Figure 11.

As shown in Figure 11, the airflow was slowly supplied into the inside of the lung tissue with the time period of about 1.0 s over in the case of the rat, and kept for a few seconds; then, the pressurized air in the lung tissue was released into the atmosphere. The air exhaust was driven by the elasticity of the lung tissue, and thus the airflow was gradually decreased to zero over a time period of about 1.0 s by the lung elasticity. It takes more time if the lung tissue becomes stiff by the disease.

On another front, the pressure value gradually increased according to the airflow supply by the syringe, and it maximized after the airflow supply. However, it slightly decreased and then became a constant, which was different from the results in the case of the balloon. Generally, the living tissue has both the elastic and the viscosity properties, and thus we now guess that the viscoelastic property in the lung tissue is related to the reason for this pressure value decrease, and the pressure in the lung tissue was balanced in a delayed way. As in the balloon experiment, the obtained relationship between the airflow volume and the pressure value in the respiratory system measured under the statically pressure-balanced condition is plotted by the open dot in Figure 12.

Next, we examined the elasticity of the lung tissue in the living rat under the oscillating airflow. The sinusoidal-shaped oscillating airflow was applied and its volume was increased in 0.5-mL increments. In this study, the ventilation frequency was set to 1.67 Hz (100 breaths/min), which is the respiratory frequency of a typical rat, and the tidal volume was changed from 0.5 to 2.5 mL.

The airflow and the pressure waveforms when the airflow volumes were 2.0 mL, 2.5 mL, and 3.0 mL in the case of the ventilator are shown in Figure 13. The sinusoidal-shaped airflow was mechanically supplied into the inside of the lung system by the ventilator by the same method as in balloon experiments, and then the stored air was expired by the lung elasticity. Thus, the airflow gradually decreased to zero in the case of the air exhaust, and its waveforms were almost coincident with those obtained in the above-stated static experiments performed by the syringe. The pressure value increased with the increase of the applied airflow volume. The pressure value, in which the airflow direction was changed from the supply to the exhaust, was used as the value under the quasi-static condition. The relationship between the airflow volume and the pressure value in the lung system in the living rat obtained under the sinusoidal-shaped oscillating airflow condition is plotted by a solid black circle in Figure 12.

As we can see, the pressure value linearly increased with the increase of airflow volume up to 3.0 mL. Generally, the differential pressure value in the respiratory system, and the critical differential pressure value at which damage of the respiratory system occurs, are 1.0–2.0 kPa and 3.4–5.9 kPa, respectively. However, we found that the maximum differential pressure between the respiratory inside and the atmosphere in our experiments was 0.35 kPa at the absolute pressure value of 100.15 kPa. Thus, we conclude that the lung tissue was normally pressurized without damage. These results indicate the following:(1)The variation of the pressure value by the airflow volume depends on the lung tissue elasticity, which demonstrates that we are able to estimate its elasticity by the gradient of the pressure vs. flow volume curve. In the case of our lung system experiment, the curve showed the linear relation in the pressure range from 100.15 kPa to 100.50 kPa (pressure difference: 0.35 kPa), and thus we conclude that the elasticity of the lung tissue is constant in this pressure region.(2)The experimental results obtained under both the static and quasi-static conditions are generally coincident. This result followed the same trend as the results of the balloon experiment. Therefore, we demonstrated that the developed respiratory measurement system is able to evaluate the elasticity of the lung tissue in a living rat by using the pressure value under the quasi-static condition in the case of ventilation (animal experiments).

## 4. Conclusions

In this work, we newly integrated both airflow and pressure sensors into our respiratory measurement system to minimize the artificial dead space in the ventilation during animal experiments. A hot-wire thermal airflow meter with a directional detection and airflow temperature change compensation function and a commercially available pressure sensor produced by MEMS technologies were assembled onto a chip sized 26.0 mm × 15.0 mm, and the dead space was reduced to the value of 0.11 mL, which is just 23% of that in the previous device. We also evaluated the elasticity of the lung tissue in a living rat under the oscillating airflow and obtained the following results: (1)The elasticity of the lung tissue in a living rat can be evaluated by gradient of the pressure vs. flow volume curve.(2)It can be also evaluated by using the pressure value obtained under the quasi-static condition in the case of the sinusoidal-shaped oscillating airflow ventilation used in the animal experiments.

We now think that the developed measurement system composed of both airflow and pressure sensors has potential, which will be used in the following different applications in animal experiments.

(1)The system can evaluate the respiration properties in the case of the spontaneous breathing by simply connecting it to the airway of the experimental animal (under no ventilation condition).(2)By integrating a CO_2_ sensor working as a capnometer onto the system, it can detect the value of its partial pressure in the breathing, and thus we can estimate the respiration conditions, such as the arrested respiration, and the partial CO_2_ pressure value in arterial blood.

The system will also be applied to a bag-valve mask, and Jackson Rees used for resuscitation in medical emergency in the case of humans, because of its miniaturization and weight-saving potential. The structure of the system will be redesigned, in which the Reynolds number becomes the same.

## Figures and Tables

**Figure 1 sensors-21-05123-f001:**
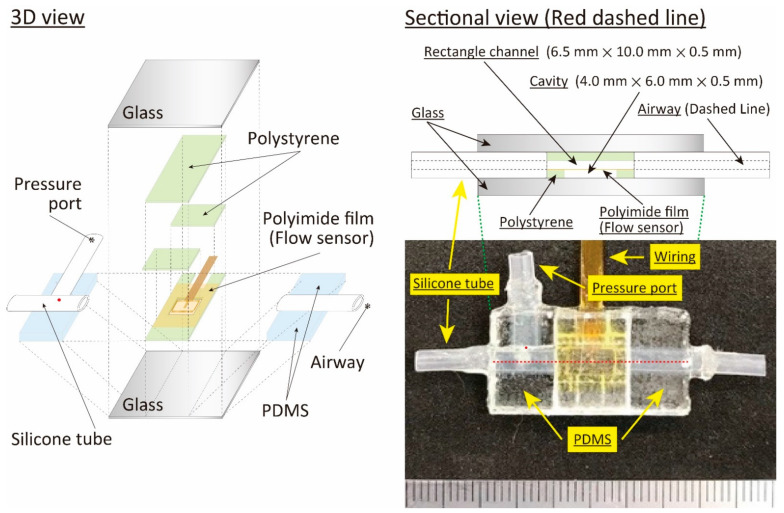
Structure of the designed respiratory measurement system for the animal ventilator.

**Figure 2 sensors-21-05123-f002:**
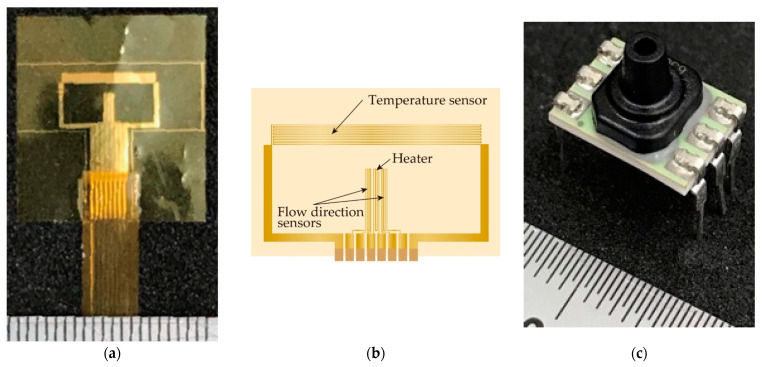
Thermal airflow and pressure sensors used in respiratory measurement system: (**a**) fabricated sensor film; (**b**) pattern of hot-wire thermal airflow meter; (**c**) pressure sensor used in this study.

**Figure 3 sensors-21-05123-f003:**
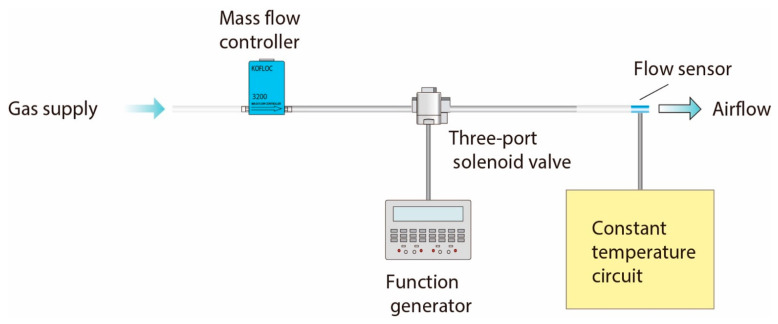
Experimental setup for airflow sensor calibration.

**Figure 4 sensors-21-05123-f004:**
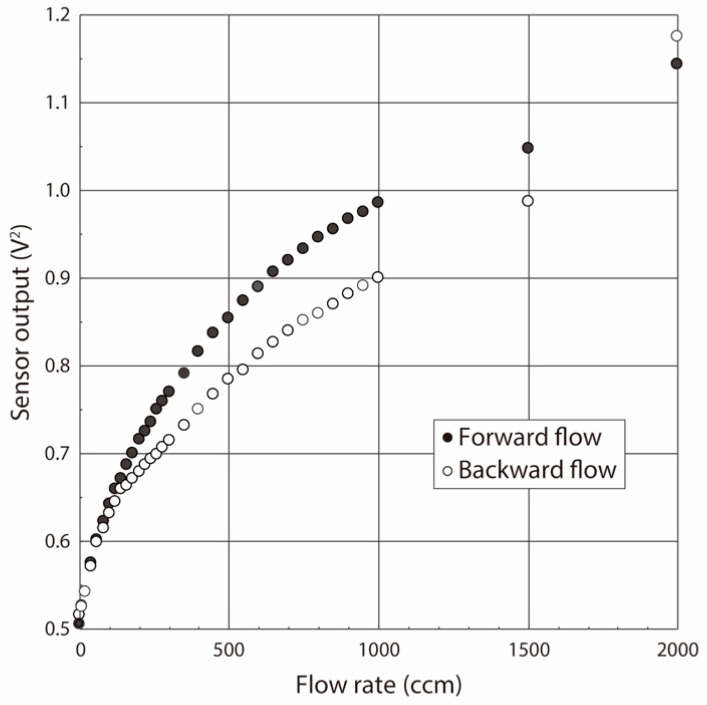
**C**alibration curve of flow sensor under both forward and backward flow directions.

**Figure 5 sensors-21-05123-f005:**
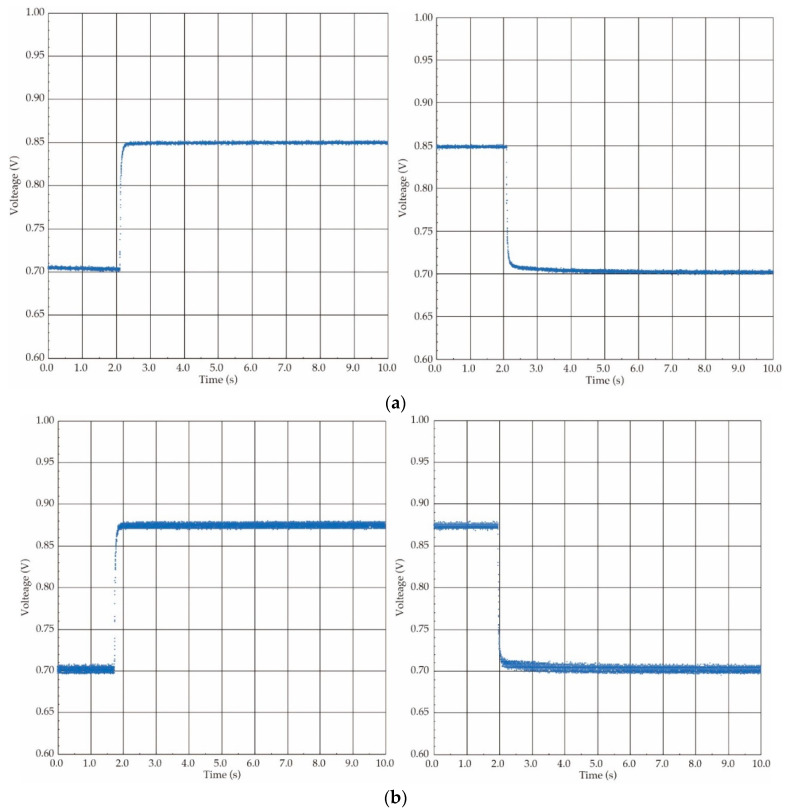
Response waveform of the airflow sensor (flow rate: 300 ccm): (**a**) forward flow direction; (**b**) backward flow direction.

**Figure 6 sensors-21-05123-f006:**
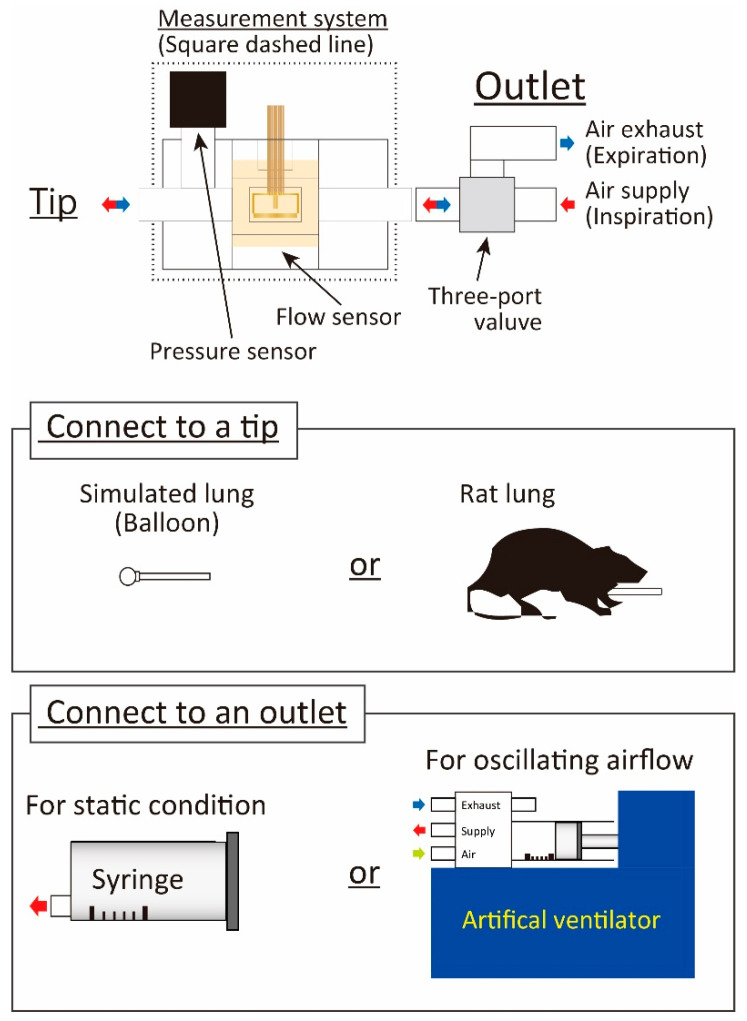
Experimental setup.

**Figure 7 sensors-21-05123-f007:**
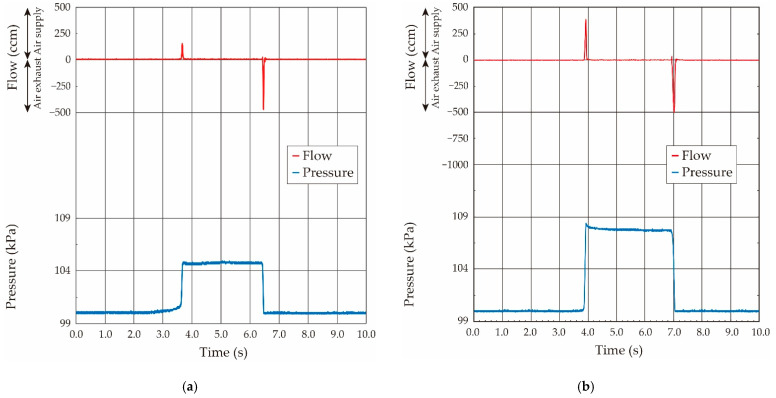
Measured both airflow and pressure waveforms when airflow volume was supplied by syringe: (**a**) airflow volume: 0.2 mL; (**b**) airflow volume: 0.6 mL.

**Figure 8 sensors-21-05123-f008:**
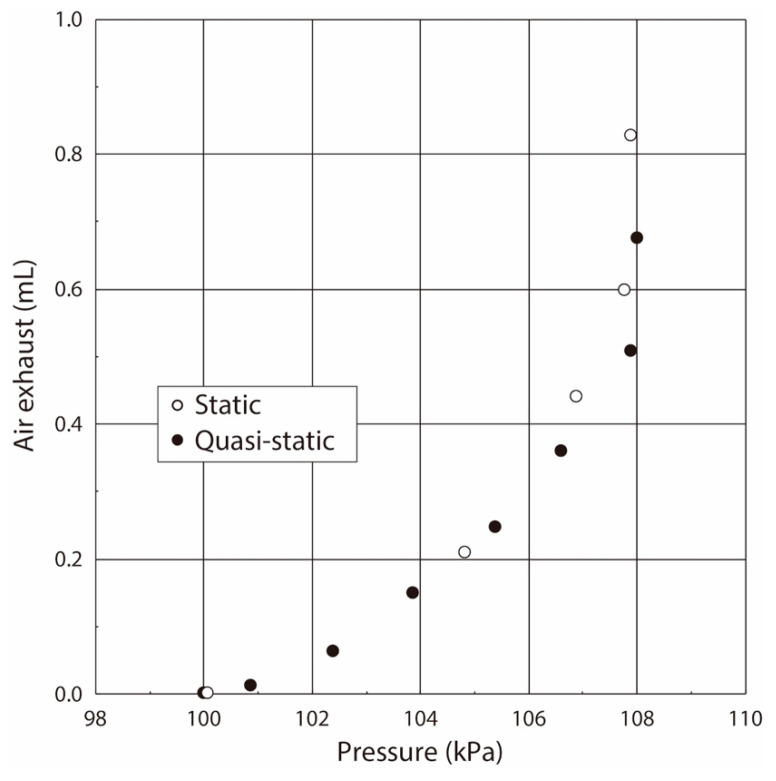
Relationship between pressure and airflow volume inside the balloon.

**Figure 9 sensors-21-05123-f009:**
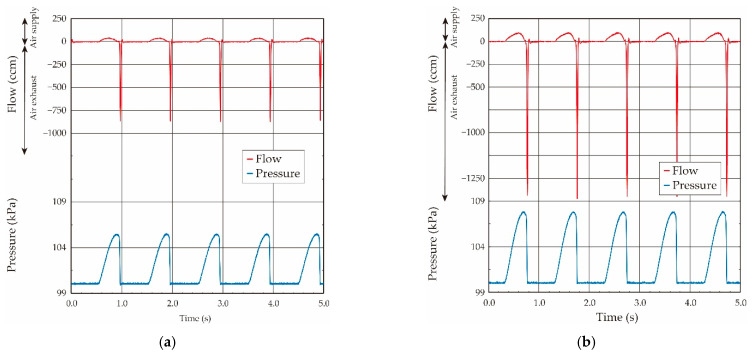
Both measured airflow and pressure waveforms when sinusoidal shaped oscillating airflow was supplied by the ventilator: (**a**) airflow volume: 1.0 mL; (**b**) airflow volume: 1.5 mL.

**Figure 10 sensors-21-05123-f010:**
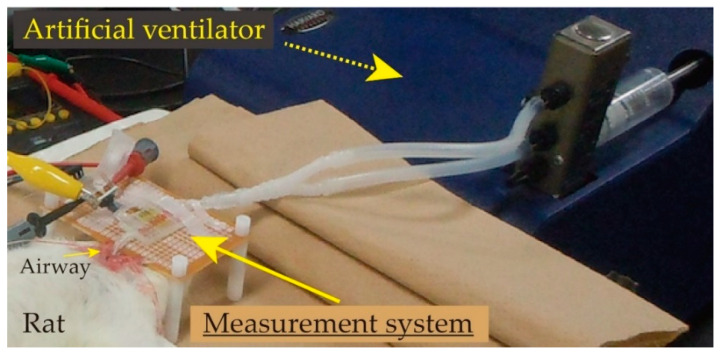
Photograph of the experimental setup in the case of animal experiment (ventilator was connected).

**Figure 11 sensors-21-05123-f011:**
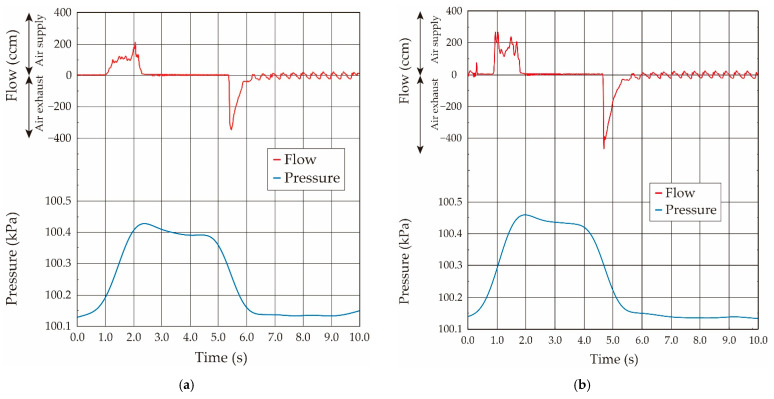
Both measured airflow and pressure waveforms when the airflow volume was supplied by the syringe: (**a**) airflow volume: 2.0 mL; (**b**) airflow volume: 2.5 mL; (**c**) airflow volume: 3.0 mL.

**Figure 12 sensors-21-05123-f012:**
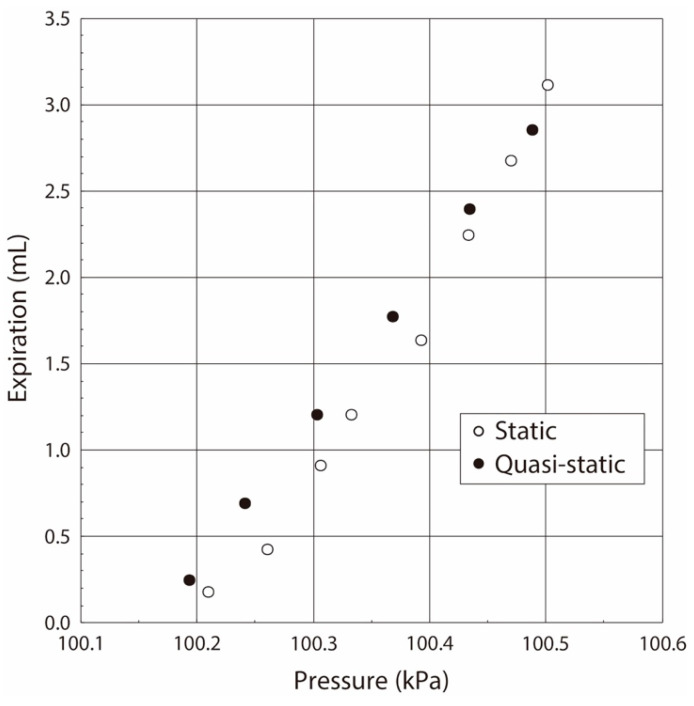
Relationship between pressure and airflow volume in respiratory interior.

**Figure 13 sensors-21-05123-f013:**
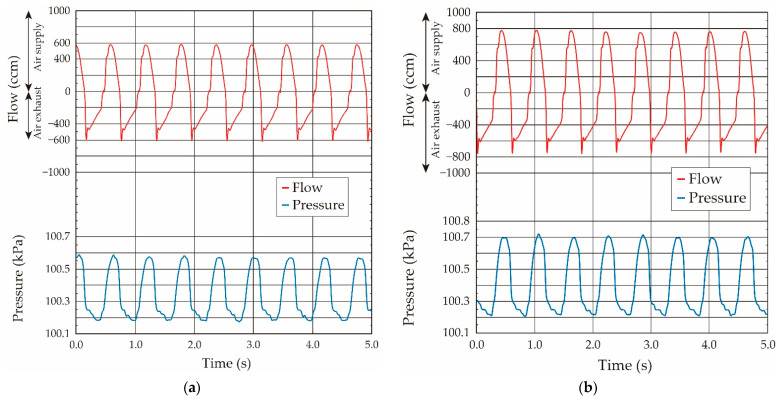
Both measured airflow and pressure waveforms when the sinusoidal-shaped oscillating airflow was supplied by the ventilator: (**a**) airflow volume: 2.0 mL; (**b**) airflow volume: 2.5 mL; (**c**) airflow volume: 3.0 mL.

## Data Availability

No new data were created or analyzed in this study. Data sharing is not applicable to this article.

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
