# Peer review of "Miniaturization of Respiratory Measurement System in Artificial Ventilator for Small Animal Experiments to Reduce Dead Space and Its Application to Lung Elasticity Evaluation"

_sensors, 2021, doi:10.3390/s21155123_

Round 1

Reviewer 1 Report

This paper investigates an interesting approach to miniaturize and adapt known sensor technologies to a new application within biomedical technologies. The paper is well structured and describe the methodology and results in sufficient details. I would recommend for publication in Sensors after some minor corrections:

  • Some reflections on how the system could be applied on a new biomedical device are missing. If that should be used in humans at some point, which kind of modifications would be required and how that would affect the device performance?
  • In page 4, line 131, the author mentioned the theoretical Reynolds number... how is this calculated, and there is any way to verify that statement experimentally?
  • I have found some typos and minor language errors, for example in page 3, line 115 (time of flight), or page 4, line 123 (two layer metals... were....). A final review for typos might be necessary

Otherwise, it is an interesting paper with convincing results.

Reviewer 2 Report

In this paper, a hot-wire thermal airflow meter and a commercial pressure sensor by MEMS technologies were used to minimize the artificial dead space in the ventilation during the animal experiment. Furthermore, the elasticity of the artificial lung tissue in a living rat under the oscillating airflow was evaluated systematically. The reviewer thinks that results obtained by the authors, such as the elevation of the lung tissue elasticity by the gradient of the pressure vs. flow volume curve and the pressure value obtained under the quasi-static condition in the case of the sin-shaped airflow ventilation, are useful for other researchers in this field. The manuscript is also well written. So, it is recommended for the acceptance. Some suggestions can be considered before the publication as follows.

  1. what is the effect of the artificial dead space reduction due to the MEMS technologies in the authors’ experiments? This point may be stated more clearly in the expression.
  2. In section 2, lots of words were used to introduce the hot-wire thermal airflow meter. Although the pressure sensor is a commercial one, It is suggested that a little bit more parameters should be introduced, such as the critical size, the reason for choosing this model, etc.. Furthermore, it is better to introduce the airflow meter and the pressure sensor separately in different paragraph.
